# The Gut Microbiome in Early Life Stress: A Systematic Review

**DOI:** 10.3390/nu15112566

**Published:** 2023-05-30

**Authors:** Ana Agusti, Femke Lamers, Maria Tamayo, Carlos Benito-Amat, Gara V. Molina-Mendoza, Brenda W. J. H. Penninx, Yolanda Sanz

**Affiliations:** 1Microbiome, Nutrition & Health Research Unit, Institute of Agrochemistry and Food Technology, Excellence Center Severo Ochoa-Spanish National Research Council (IATA-CSIC), 46980 Valencia, Spain; mtamayo@iata.csic.es (M.T.); gara.molina@iata.csic.es (G.V.M.-M.); 2Amsterdam UMC, Amsterdam Public Health, Mental Health Program, Department of Psychiatry, Vrije Universiteit Amsterdam, Boelelaan 1117, 1081 HV Amsterdam, The Netherlands; f.lamers@amsterdamumc.nl (F.L.); b.penninx@vumc.nl (B.W.J.H.P.); 3Department of Medicine, Autonomous University of Madrid, 28029 Madrid, Spain; 4Institute for the Management and Innovation of Knowledge (INGENIO-CSIC-UPV), Polytechnic University of Valencia, 46022 Valencia, Spain; cbenito@ingenio.upv.es

**Keywords:** gut microbiome, gut-brain axis, early life stress, prenatal stress, postnatal stress, systematic review

## Abstract

Exposure to early life stress (ELS), prenatal or postnatal during childhood and adolescence, can significantly impact mental and physical health. The role of the intestinal microbiome in human health, and particularly mental health, is becoming increasingly evident. This systematic review aims to summarize the clinical data evaluating the effect of ELS on the human intestinal microbiome. The systematic review (CRD42022351092) was performed following PRISMA guidelines, with ELS considered as exposure to psychological stressors prenatally and during early life (childhood and adolescence). Thirteen articles met all inclusion criteria, and all studies reviewed found a link between ELS and the gut microbiome in both prenatal and postnatal periods. However, we failed to find consensus microbiome signatures associated with pre- or postnatal stress, or both. The inconsistency of results is likely attributed to various factors such as different experimental designs, ages examined, questionnaires, timing of sample collection and analysis methods, small population sizes, and the type of stressors. Additional studies using similar stressors and validated stress measures, as well as higher-resolution microbiome analytical approaches, are needed to draw definitive conclusions about the links between stress and the human gut microbiome.

## 1. Introduction

Early life stress (ELS) is associated with myriad negative neuropsychiatric and physical health outcomes in later life. The concept of ELS is broad and includes both pre- and postnatal stress. Prenatal stress (or prenatal maternal stress) refers to any psychological or physical stress experienced by the mother during pregnancy that impacts fetal health [1]. Postnatal stress is related to any psychological/physical stress, including emotional, physical and sexual, childhood neglect, parental psychopathology and separation, and adolescent bullying, victimization, or violence [2]. Children exposed to ELS show alterations in brain development and are at increased risk of developing mental illness [3]. Indeed, it has been suggested that an adverse environment may contribute up to 45% of the development of mental illness in children and up to 30% in adults [4,5].

Chronic psychological stress is associated with the dysregulation of the hypothalamic-pituitary-adrenal (HPA) axis, manifesting as elevated cortisol levels and alterations in the immune system and the gut microbiome [6,7]. While the acute stress response is regulated through physiological changes in the endocrine and nervous system, ultimately returning to the basal state, chronic and/or severe stress, particularly during childhood, can lead to long-lasting activation of stress pathways. This has detrimental physical and psychological consequences for major biological systems, including metabolic, cardiovascular, endocrine, immune, and nervous systems [8,9]. Many studies have investigated the mechanisms linking ELS to these systems and, more recently, several have examined the relationship between stress and the gut microbiome specifically in early life [10,11]. Some of these human studies, which are included in this systematic review, are summarized in Figure 1.

The gut microbiome is a complex ecosystem consisting of trillions of microorganisms including bacteria (representing the majority), viruses, protozoa, helminths, fungi, and archaea [24]. The term microbiome refers to the collective genome and activity associated with a specific host habitat or environment. The gut microbiome is involved in nutrient and xenobiotic metabolism, and in the development and function of the endocrine and immune systems and the gut barrier [25,26]. The gut microbiome participates in bidirectional communication with the brain to modulate central nervous system function [27], and preclinical studies indicate that gut microbes can influence brain behavior [28,29]. Alterations in the gut microbiome have been linked to mental health conditions such as depression or anxiety in pre-clinical and clinical studies [29,30,31,32].

The colonization and development of the gut microbiome is critical during the first years of life and may have long-lasting consequences owing to its intimate interaction with the immune system [33]. While there is debate over when the first contact with microbes begins, pre- or postnatally, it is clear that massive colonization of the infant gut by different microorganisms starts at birth. The extent of this is contingent on several factors such as the mode of delivery (vaginal or cesarean), prematurity, antibiotic treatments, feeding practices (breastfeeding or formula feeding), introduction of complementary food, exposure to animals (pets), number of siblings, or psychological stress [3,16]. The development of the gut microbiome from childhood to adulthood rests mainly on lifestyle and diet factors [16] and is characterized by an increase in the diversity of microbial species. Adolescence is also a critical developmental period when the individual is exposed to many challenges and stressors and the brain is developing. Accordingly, the richness of the gut microbiome also changes during this period, which can influence brain development and function [34].

Recent studies have shown that maternal prenatal stress is associated with infant development outcomes on several levels, emotional, behavioral, and cognitive [35,36], and with increased risk of developing somatic conditions such as asthma [37]. Indeed, even moderate stress exposure may have an influence if it occurs chronically [38]. Although the precise molecular mechanisms underlying the adverse effects of prenatal stress on offspring remain enigmatic, studies in animal models suggest that it can impact the microbiome [39,40]. Moreover, the few studies in humans point to a role for the gut microbiome in mediating prenatal stress and associated outcomes [12,13,14,15]. Hantsoo and Zemel [41] recently reviewed the impact of ELS on the gut microbiome and the role of dietary interventions to moderate its impact, but not systematically. The present systematic review aims to compile and discuss the existing scientific evidence on the link between ELS (vs. no ELS) and changes to the human gut microbiome. We sought to clarify whether ELS influences the gut microbiome and whether this can be a biological predictor and/or causative factor for the development of somatic or mental disorders. We review the literature evaluating how ELS shapes the gut microbiome, especially pre- and postnatal stress.

## 2. Materials and Methods

### 2.1. Search Strategy and Inclusion/Exclusion Criteria

The systematic review was conducted following the Preferred Reporting Items for Systematic Reviews and Meta-Analysis (PRISMA) guidelines [42]. The protocol was registered in PROSPERO under the Registration Number CRD42022351092.

We searched for published articles in Medline (Pubmed), EMBASE, Web of Science and Scopus (via Elsevier), in which the title, abstract, and keywords contained terms related with child abuse, adverse child experiences, maternal separation, ELS, gut microbiome, and human. Searches were performed on 30 April 2022 using a combination of keywords of subject and free text terms, with no date limit or language restriction. The strategy was developed for Medline and then adapted for other databases. The complete Medline strategy, including keywords, is shown in Appendix A.

Eligibility criteria included reports comparing ELS versus no ELS on gut microbiome outcomes in humans in the following types of study: case report, clinical trial, clinical trial protocol, clinical trial phase I–IV, controlled clinical trial, evaluation study, multicenter study, observational study, twin studies, and validation study. In vitro and animal studies were excluded. Different exposures and/or interventions were reviewed including prenatal exposure, such as in utero exposure to clinically significant depression, exposure to precarious situations, or serious illness during pregnancy; and postnatal exposure, such as emotional, physical, and sexual abuse or neglect in childhood, parental psychopathology and separation, prepubertal bullying, in addition to victimization or violence and serious physical illness during childhood or adolescence. 

Titles, abstracts, and full texts of articles were screened independently by two reviewers (Agusti, A. and Lamers, F.) for eligibility using Abstract r free software (Brown University, Providence, RI, USA, http://abstrackr.cebm.brown.edu, accessed on 30 April 2022). Disagreement was resolved by discussion and by a third senior reviewer (Sanz, Y.) when needed.

### 2.2. Data Extraction

Data extracted included: (a) study data (journal, authors, publication year, study design, type of sample collected (feces), methods of gut microbiome analysis, measures of ELS and methods of assessing ELS, (b) sample description (sex, age, and ethnicity), and (c) effect sizes (analyses used to compare the gut microbiome between cases and controls, or association studies between ELS and gut microbiome changes). Data were extracted by two independent authors (Lamers, F. and Agusti, A.). 

### 2.3. Assessment of Risk of Bias

Quality assessment of risk of bias of non-randomized studies, including cohort studies and case-control studies, was performed with the Newcastle-Ottawa Scale (NOS) [43], which follows the “star system” and evaluates 8 items grouped into 3 categories: selection of participants (maximum 4 stars), comparability of the groups (maximum 2 stars) and ascertainment of the outcomes (for cohort studies) or exposure (for case-control studies) of interest (maximum 3 stars). For quality assessment of risk of bias of cross-sectional studies, we used an adapted NOS scale created by Herzog et al. [44], which evaluates 7 items grouped into the same 3 categories: participants (maximum 5), comparability (maximum 2), and ascertainment of the outcomes (maximum score 3). Each study can be awarded a maximum of 10 stars, with a higher score indicating better methodology quality. Two authors (Agusti, A., Tamayo, M.) independently assessed the risk of bias of individual studies and any differences were resolved through consensus.

### 2.4. Strategy for Data Synthesis and Analysis of Subgroups

Because study designs and outcome assessments varied, the results are presented in a narrative manner with tables. Studies are presented based on the population, intervention (or exposure), comparison (if applicable), outcome criteria. We considered the subgroups pre- and postnatal stress.

## 3. Results

### 3.1. Overview 

Of 202 articles found, 50 were duplicates and were removed before screening. Subsequently, 132 articles were excluded based on title/abstract. Seven of the remaining 20 full-text articles assessed for eligibility were also excluded, resulting in a total of 13 studies included. Figure 2 describes the screening and selection process in full. Studies were grouped into two categories: prenatal and postnatal stress. All 13 studies were observational. The four prenatal studies were prospective in design [12,13,14,15]. The nine postnatal studies included one longitudinal study [19], five case-control studies [16,17,18,20,23], and three cross-sectional studies [11,21,22]. All 13 studies were published between 2015 and 2022. Geographically, the distribution of the studies was: five in the U.S.A. [11,16,18,19,21], five in Europe [12,14,17,20,22], two in Africa [15,23], and one in South America [13].

Prenatal stress included not only studies in which the pregnant mother experienced psychological stress (depression and/or anxiety [12,14]) but also in which the mother’s lifestyle and physical circumstances may have caused stress to the baby, such as being exposed to precarious situations [13,15] or suffering from a serious illness such as HIV [15]. The postnatal studies included children and/or adolescents experiencing ELS; for example, studies on children/adolescents in institutional care [16,17,18], a study on adults with post-traumatic stress disorder (PTSD) in whom the impact of stress during their childhood had been assessed [23], a case-control study evaluating the impact of maternal prenatal psychological stress on the stress response of babies [20] and finally, studies investigating ELS reported by parents via questionnaire [11,19,21], sex-specific associations, and preschool-age neurobehavior [19].

### 3.2. Risk of Bias 

#### 3.2.1. Quality Assessment of Longitudinal Cohort Studies Based on the Newcastle-Ottawa Scale

Five longitudinal studies were evaluated following NOS methodology [43] (Table 1). In the category of “selection”, two studies received a rating of three because one used a self-report questionnaire rather than a structured interview, failing in the “ascertainment of the exposure”, and the other study failed to report information on the “outcome of interest at start of study”. One study received a rating of two, as information was only collected with a self-report questionnaire and no information was provided on “outcome of interest at start of study”. Two studies received a maximum score of four as they satisfied all the conditions in the selection of participants for a high-quality longitudinal study. In the “comparability” category, one study received a rating of one because it did not include appropriate confounders in the analysis. Five studies were awarded a rating of four because they included different confounders in the adjustment of the analysis, such as peripartum antibiotic exposure, exclusively or mixed breastfed, vaginal delivery, infant sex, or infant age during sampling. In the “outcome” category, only one study achieved the maximum score of three because it met all stipulated requirements. Four studies received two stars because they did not specify the “adequacy of follow-up of cohorts”. Four studies received a total score of seven [12,13,14,15] and one study had a total of eight [38].

#### 3.2.2. Quality Assessment of Case-Control Studies Based on the Newcastle-Ottawa Scale

Five case-control studies were evaluated based on NOS methodology [43] (Table 2). In the category of “selection”, three studies received the maximum score of four because all adequately defined cases with good representativeness and suitable selection and definition of controls. The two remaining studies received three stars because none of them reported details about the participants (e.g., where they were recruited, hospital, clinic, city); therefore, the “representativeness of the cases” was rated 0. In the “comparability” category, one study received a rating of one because it did not adjust the analysis for appropriate confounders. Contrastingly, the remaining four studies included several confounders in the analysis, including birth mode, antibiotics, age, presence of siblings, country of origin, sex, diet, breastfeeding, and ethnicity, and received the maximum score of two. In the “exposure” category, all five studies received a rating of two. All used a diagnosis or a structured interview to assess the exposure and used the same method for control participants. Moreover, none of the studies clearly described the non-response rate in each group. In sum, three of the five studies received a final rating of seven [16,18,20] and two received a rating of eight [17,23]. 

#### 3.2.3. Quality Assessment of Cross-Sectional Studies Based on the Newcastle-Ottawa Scale

Three cross-sectional studies [11,21,22] were evaluated with the adapted NOS [44] (Table 3). One study received a score of two in the “selection” category, as it failed to specify the selection of the control group, only used self-report questionnaires to evaluate the stressor, and provided no data about non-response. The second study received a rating of three, as it failed to report on non-response, and the third study received four stars. In the “comparability” category, one study received one star as it only included one confounder (diet) in the adjustment of the analysis. The remaining two studies received the maximum score of two as they included several confounders in the analysis, such as age, gender, socio-economic status, food intake frequency of fiber-rich food, protein-rich food, sweet food and fatty food, and day sleep duration. The last category evaluated was “outcome”, and all three studies received the maximum score of three. In sum, one study was rated six [21], one was rated eight [22], and the third study was rated nine [11]. 

### 3.3. Prenatal Stress 

We reviewed the four human longitudinal studies published at the time the bibliographic search was conducted (Appendix A).

#### Longitudinal Studies

In a large prospective study in Finland (*n* = 446) [12], maternal prenatal psychological distress and hair cortisol levels were both associated with changes in the gut microbiome of 2.5-month-old infants. Chronic maternal prenatal psychological distress evaluated by different standardized questionnaires was negatively associated with *Akkermansia*, *Phascolarctobacterium*, and *Megamonas* abundance and positively with *Veillonella*, *Finegoldia*, *Dialister*, *Dorea*, and *Coprococcus* abundance. Likewise, maternal hair cortisol levels were related to the infant gut microbiome, and were negatively associated with *Lactobacillus* (phylum *Firmicutes*), *Slackia* and *Actinobaculum* (phylum *Actinobacteria*), *Butyricimonas* (phylum *Bacteroidetes*), and *Citrobacter* (phylum *Proteobacteria*). The second prospective longitudinal cohort study compared 272 mother–infant pairs from Nigeria [15] during the first 18 months of life exposed to maternal HIV infection but not infected themselves versus infants unexposed to maternal HIV. We assumed that experiencing a serious illness such as HIV generates physical and possibly psychological stress in the mother. Although a direct relationship between maternal HIV and the infant microbiome was not reported, antiretroviral drugs in the breast milk of HIV mothers was associated with a lower relative abundance of *Bifidobacterium longum* in non-infected infants. The third longitudinal cohort study, performed in 25 mother–infant dyads, revealed associations between exposure to precarity and alterations in the HPA axis during peripartum, and alterations in the gut microbiome of the offspring [13]. Measures of maternal precarity were obtained during and after pregnancy using validated questionnaires, and the HPA axis was evaluated through cortisol measurements in saliva from the mother during/after pregnancy. Saliva and stool samples were also obtained from the newborns at 3 days and 2 months of life. The authors reported that both measures of precarity exposure and HPA dysregulation were consistently associated with gut microbiome alterations in the infants, characterized by decreased species diversity and abundance of *Bifidobacterium* at the genus level and increased abundance of potential pathogens of the family *Enterobacteriaceae.* In the final prospective longitudinal study in Dutch children followed from the 3rd trimester of pregnancy until 110 days after birth [14], the authors found that prenatal stress assessed by questionnaires or by measures of basal cortisol in saliva, or both, was associated with the gut microbiome composition of the offspring during at least 3 months. Infants with high cumulative prenatal stress (high reported stress and high cortisol) had a higher relative abundance of Proteobacteria (*Escherichia*, *Serratia*, and *Enterobacter*), lower relative abundance of lactic acid bacteria (*Lactobacillus*, *Lactoccus*, and *Aerococcus)* and Actinobacteria (*Bifidobacterium*, *Collinsella*, and *Eggerthella*) and more gastrointestinal symptoms and allergic reactions reported by the mothers, suggesting an association with the dysbiosis. 

In sum, although the four prenatal studies reported different results, there are some similarities; for example, in the association of stress symptoms/conditions with a lower abundance of *Bifidobacterium* and increased abundance of the *Enterobacter* genus (see Table 4 for additional information).

### 3.4. Postnatal Stress

#### 3.4.1. Longitudinal Studies

Laue et al. [38] designed a 3-year follow-up prospective study to test for sex-specific associations between the gut microbiome and behavior, and collected stool samples from 260 children at 6 weeks, 1 year, and 2 years postpartum. When the children were ~3 years old, their parents completed a behavior assessment questionnaire to capture different types of phenotypes. Results showed that most outcomes were not related to beta diversity changes; however, higher microbiome diversity at the youngest age (6 weeks) was related to lower depression in the overall sample and with lower anxiety and better internalizing behavior, especially in boys. Specifically, better composite scores for adaptive skills in boys associated positively with *Bifidobacterium* abundance and negatively with *Klebsiella* abundance. In girls, *Granulicatella* was associated with worse anxiety scores at 6 weeks of age, whereas at 1 year *Streptocuccus peroris* was associated with better internalizing problems. Finally, some *Blautia* species were linked to worse hyperactivity scores with stronger associations among girls (Table 5). 

#### 3.4.2. Case-Control Studies

One study of 344 children aged 3–18 years investigated whether early adversity (EA), specifically parental deprivation, international adoption, or institutional care, was related to gastrointestinal and mental disorders and to changes in microbiome diversity (115 with EA and 229 controls) [16]. A positive association was found between EA experiences and gastrointestinal symptoms and anxiety. A sub-sample of children was then used to analyze the microbiome in stool samples and its relationship with functional magnetic resonance imaging of the brain (5–11 years old). Despite the low sample size (N = 8 EA; N = 8 control), the authors found changes in alpha (richness) and beta (uniqueness) diversity as well as decreases in Lachnospiraceae and an unknown bacterium in the EA group compared with controls, which associated with brain reactivity within emotion networks and with the frequency of diarrhea. A second case-control study performed by Malan-Muller et al. [23] investigated the associations between the gut microbiome and mental health outcomes in adults with PTSD and trauma-exposed controls (TE controls). The authors also reported on several questionnaires to evaluate potentially traumatic lifetime events, major depressive disorder (MDD) and anxiety disorders. A fecal sample was collected within the same week as the clinical assessment. The results showed that a consortium of four genera (*Mitsuokella* and *Odoribacter*, *Catenibacterium*, *Olsenella*) was positively associated with PTSD status and MDD was associated with a higher relative abundance of the phylum *Bacteroidetes*. Individuals using psychotropic medication (PTSD and TE controls) at the time of sampling showed an increase in the relative abundance of *Ruminococcus* and a decrease in *Akkermansia*. Treatment was also positively associated with *Bacteroidetes*, *Firmicutes*, and negatively with *Verrucomicrobia*. Keskitalo et al. [20] studied the link between gut microbiome composition and the cortisol releasing response to a stressor. Children from the FinnBrain Birth Cohort with an available fecal microbiome profile and a salivary test of cortisol at 2.5 months of age were included in this nested case-control study. Cortisol levels were evaluated 0, 15, 25, and 35 min after exposing the children to a mild acute stressor. Results showed that a blunted cortisol stress response was weakly associated with gut microbiome diversity, but associations were found between cortisol levels and the taxonomic composition of the fecal microbiome. Another case-control study associated ELS in adolescents adopted internationally from orphanages into the United States [18] with alterations in gut microbiome and inflammatory markers in a case-control study as compared with youth reared in birth families (controls). Results showed that the abundance of several bacterial taxa was significantly higher in the stressed group than in the control group, including the genera *Prevotella*, *Bacteroides* (Bacteroidaceae), *Coprococcus*, *Streptococcus*, and *Escherichia.* Cytomegalovirus was also higher in the stressed group and there were also differences in bacterial abundance in viral seropositive groups compared with seronegative groups: *Escherichia*, Ruminococcaceae, Lachnospiraceae, and Catabacteriaceae. Moreover, *Escherichia* was related to seropositive individuals in the stressed group (see Table 5 for additional information).

#### 3.4.3. Cross-Sectional Studies

A cross-sectional study [11] explored the relationship between the gut microbiome, its metabolites, and brain alterations in adults with ELA, measured by the Early Traumatic Inventory-Self Report (ETI-SR), which includes 27 questions about general trauma, physical punishment, emotional abuse, and sexual abuse. The study enrolled 128 adults without psychiatric conditions as assessed by the modified MINI questionnaire. They also collected stool samples and performed cerebral structural and functional magnetic resonance. Results showed that higher scores in ETI-SR (>4) versus lower scores (≥4) were not associated with differences in microbial diversity or abundance of specific taxa. They also found that symptoms of anxiety, depression, and body mass index correlated significantly with several feces metabolites such as urate, glutamate gamma-methyl ester, and 5-oxoproline. These scales as well as current stress related significantly to brain functional connectivity of sensorimotor, central executive, default mode, and central autonomic regions, and subsets of these networks, in addition to salience, emotion regulation and occipital correlated significantly with the metabolites. A cross-sectional study performed by Michels et al. [22] with children and adolescents investigated the link between the gut microbiome in feces and psychosocial stress. Stress was reported with different questionnaires assessing negative events, negative versus positive emotions and emotional problems. Moreover, cortisol was measured as a stress biomarker and heart rate variability (pnn50) was used as a measure of parasympathetic nervous system activity. The results showed that high levels of stress (low pnn50 and elevated negative events) were associated with a decrease in alpha diversity. Adjusted and unadjusted taxonomic differences were also more pronounced for happiness and pnn50 (as a measure of parasympathetic nervous system activity), being associated with two abundant observed taxonomic units (OTUs), respectively (24 OTUs representing 11.8% of bacterial counts and 31 OTUs representing 13.0%). Overall, high stress was related to higher levels of *Bacteroides*, *Parabacteroides*, *Rhodococcus*, *Methanobrevibacter*, and *Roseburia* but lower *Phascolarctobacterium* at genus level as well as with lower Firmicutes at the phylum level. However, conflicting results were reported between different stress measures as well as differences between preadolescents and adolescents. In another cross-sectional study [21], the role of the psychosocial environment during childhood and caregivers behavior on the gut microbiome was evaluated in 40 early school-age (5–7 years) children in the Pacific Northwest of the U.S.A. Stool samples were analyzed (16S and metagenomics) in addition to a wide range of socioeconomic factors (socioeconomic risk, behavioral dysregulation, caregiver behavior, demography, gut-related history, and diet). Specifically, the microbial taxon that varied with behavioral disturbances was *Bacteroides fragilis*, which was associated with reduced levels of aggressivity, emotional reactivity, sadness, and impulsivity, as well as lower family incidents. Contrastingly, two butyrate-producing bacteria (*Coprococcus comes* and *Eubacterium rectale*) were associated with more anxious and depressive problems and less inhibitory control. However, *Roseburia inulinivorans*, also a butyrate-producing bacterium, was associated with a decrease in depressive problems. The authors further found associations between individual taxa (e.g., *B. fragilis*) and functional groups (e.g., monoamine metabolism) within the microbiome and metrics of socioeconomic risk and behavioral dysregulation. More details of case-control studies are compiled in Table 5.

## 4. Discussion

In the present systematic review, we sought to clarify the effect of ELS on the human gut microbiome by assessing the strengths and weaknesses of the existing literature. Although the knowledge on this subject is by no means extensive, with the majority of studies conducted in animal models, we compiled and critically reviewed human studies investigating both pre- and postnatal stress. Overall, the quality of the studies as assessed with the NOS scales was satisfactory, with most scoring seven or eight out of nine stars (only one study scored six). Figure 1 provides an overview of the main results of the systematic review.

The four studies on prenatal stress [12,13,14,15] were longitudinal in design and analyzed the gut microbiome at the genus level using the same sequencing platform. Three of the four studies [12,13,14] also reported maternal cortisol levels during and after pregnancy. While a detailed comparison of studies is difficult because of the different questionnaires used to assess stress, similar results were reported in some studies with respect to microbiome analysis. For example, in the Zijlmans et al. study [14], infants with high cumulative stress (high reported mother stress and high cortisol) had an increased relative abundance of Proteobacteria groups. Likewise, in the study by Janhke et al. [13], the stress perceived by pregnant mothers was associated with a high abundance of Proteobacteria in the newborn gut microbiome, in particular an unspecific genus of *Enterobacteriaceae*. The same study showed that this genus was also increased in infants with high basal cortisol levels 3 days postpartum. These results suggest that prenatal stress might be associated with an increase in the relative abundance of Proteobacteria phylum. Of note, gamma-proteobacteria have been linked to human infant necrotizing enterocolitis [45]. There were also similarities in the negative association between stress and the relative abundance of the genus *Bifidobacterium* in at least three of the four prenatal studies. Janhke et al. [13] found that the stress perceived by pregnant women in items including “postpartum depression” and “low family support in the postpartum” was negatively associated with *Bifidobacterium.* The same genus was also negatively associated with high infant basal cortisol levels at 3 days postpartum. Zijlmans et al. [14] also found that infants with high cumulative stress showed a negative association with the abundance of *Bifidobacterium* and lactic-acid bacteria (i.e., *Lactobacillus*, *Lactoccus*) and *Aerococcus*.

In their study on pregnant women with and without HIV, Grant-Beurmann et al. [15] found that breastfeeding positively associated with *Bifidobacterium* and *Collinsella* in children both unexposed and exposed to the virus. They also demonstrated that the abundance of the genus *Bifidobacterium* was significantly greater in unexposed children than in exposed children. That being said, antiretroviral therapy was associated with the reduction in the abundance of this genus. *Bifidobacterium* is associated with a variety of beneficial effects for health, with an important role in the barrier effect or regulation of immune system [46], but also in mental health-related diseases such as depression [47,48]. These results might suggest that prenatal stress is linked to a decrease in the relative abundance of *Bifidobaterium*, which theoretically may impact infant health. Nevertheless, we believe that the results are inconclusive and further studies are needed.

The 10 postnatal stress studies addressed diverse ELS stressors, including childhood trauma, psychiatric disorders, perceived stress, and care situations (institutionalized vs. not and childcare vs. home care). While exemplifying the diverse nature of ELS stressors, this scope hampered our ability to compare results. Indeed, the studies did not have the same end goal, although all met the stated requirements for ELS. For instance, in three of the nine studies, the impact of stress on the gut microbiome was not the final objective, but we included them in the analysis since they assessed whether institutional care for a prolonged period impacted the microbiome [16,17,18]. We assumed that these children would be under enormous stress; however, no cortisol measurements were conducted and no assessment of emotional state was made. All nine reviewed studies analyzed the microbiome at least at one time point, but only two studies included cortisol analysis.

We reviewed only one postnatal longitudinal stress study [38], in which the main objective was to test for sex-specific associations between gut microbiome and behavioral development. The authors reported that an increase in alpha diversity at 6 weeks was associated with lower levels of depression, anxiety, and internalizing problems and, at 2 years, with better social and adaptive skills, but only in boys. They also found associations between gut microbiome and adaptive skills behavior and hyperactivity at 6 weeks, 1 year, and 2 years, but no associations with domains related to ELS. We believe that the study was limited by the non-inclusion of dietary factors (with the exception of breastfeeding) or antibiotics as covariates. Moreover, most findings were identified in the sex-specific analyses and not in the unstratified analyses. This would suggest that future studies on gut microbiome and ELS should consider the sex influence.

With respect to the case-control studies on postnatal stress and gut microbiome, these varied widely in design, including participant age (ranging from newborns to adults) and the timing of fecal collection, which makes comparative analyses challenging. The type of ELS in the five studies also varied widely, including the stress of institutional care [16,17,18], PTSD as a possible childhood trauma [23] and ELS evaluated using questionnaires [20]. One of the five studies, which compared infants in institutional childcare from 3 months of age with home-cared peers, failed to find significant differences in the gut microbiome [17]. In two studies, no differences were found in the alpha diversity of the gut microbiome between cases and controls [18,23]. One of these studies [23] was performed within the “Shared Roots” parent study with patients with PTSD and TE controls [23]. Five years earlier, Hemmings et al. [49] analyzed the same cohort and although they also failed to find differences in alpha diversity between the PTSD and TE groups, they identified three phyla that were decreased and potentially important to classify PTSD vs. TE with an error rate of 30.7%: *Actinobacteria*, *Lentisphaerae*, and *Verrucomicrobia*. Interestingly, the relative abundance of *Actinobacteria* and *Verrucomicrobia* was associated with childhood trauma. In the study performed 5 years later by Malan-Muller et al. [23], the authors closely examined the impact of psychotropic medication on the gut microbiome, finding that medication in both groups associated positively with *Bacteroidetes*, *Firmicutes*, and negatively with *Verrucomicrobia*, in particular with its only genus *Akkermansia*. Accordingly, changes in the abundance of *Verrucomicrobia* (and *Akkermansia*) can be the result of the medication rather than PTSD, as discussed. Furthermore, the authors of the more recent study identified a consortium of four genera (*Mitsuokella*, *Odoribacter*, *Catenibacterium*, and *Olsenella*) to discriminate individuals with moderate PTSD from TE controls. We consider that this is a more robust study not only because of the increased taxonomic resolution to at least genus level, but also because medication was considered as a modulating factor, unlike the 2017 study. Because of this, we elected to include only the study of Malan-Muller et al. [23] in the review. Regarding the three studies reporting on children cared for in institutions or in foster care, followed by international adoption, one study failed to find any significant effect on the gut microbiome between cases and controls [17]; however, some interesting results were found in the other two studies. The Callahan et al. study [16] found that alpha diversity was lower in children exposed to adversity than in controls, whereas the study by Reid et al. [18] reported no differences in alpha diversity, and both studies found alterations in beta diversity. The Callahan study identified two biomarkers for those children cared for in institutions, but both belonged to unknown genera (one from the family *Lachnospiraceae* and the other was an unknown genus and family). The Reid study also identified changes in the *Lachnospiraceae* family between CMV serotypes, but not between the cases and controls. The role of *Lachnospiraceae* in the gut is contentious: it is considered as a main producer of beneficial short-chain fatty acids, but some members of the family are associated with intestinal diseases [50]. It is evident that a greater taxonomic resolution is needed to better interpret the potential biological meaning of these findings. The latter study also found differences in *Prevotella*, *Bacteroides*, *Coprococcus*, *Streptococcus*, and *Escherichia* at the genus level between cases and controls. The study by Keskitalo et al. [20] attempted to identify a link between cortisol measured in infant saliva and the gut microbiome at the age of 2.5 months, based on a previous study suggesting a positive association between gut microbiome alpha diversity and saliva cortisol reactivity in 1-month-old babies [51]. However, the authors failed to confirm the earlier findings.

Three cross-sectional studies related to postnatal stress were reviewed. The study by Coley et al. [11] assessed whether ELS-related changes in gut microbial metabolites are associated with alterations in brain connectivity and increased vulnerability to stress in adulthood, but did not obtain conclusive results. Nevertheless, the authors found associations between anxiety symptoms and fecal metabolites such as glutamate gamma-methyl ester, but they did not determine the responsible gut bacteria. A comprehensive study by Michels et al. [22] with children and adolescents evaluated both self-reported levels of stress as well as biomarkers (hair cortisol and pnn50) and their association with gut microbiome. They found that high stress was associated with a lower abundance of *Firmicutes* at the phylum level, and higher *Bacteroides*, *Parabacteroides*, *Rhodococcus*, *Methanobrevibacter*, and *Roseburia* but lower *Phascolarctobacterium* at the genus level. Some of these results are in line with the published literature. For example, *Firmicutes* is under-represented in major depression [52]. Nevertheless, the associations at this high taxonomic level (phylum) are of limited value. Finally, the Flannery et al. [21] study focused more on the triad of caregiver behavior, social risks, and child behavior, finding an association between *Bacteroides fragilis* and less aggressiveness, sadness, impulsivity, and lower family incidents, and an association between *Eubacterium rectale* with the increase in anxious depression. These findings contrast with Michels et al. [22], who associated the increased abundance of the genus *Bacteroides* with high levels of stress, and *Eubacterium coprostanoligenes* was associated with stress albeit with inconsistent results. Moreover, both *Eubacterium* species differ and, therefore, the findings are not consistent. Flannery et al. [21] also identified two butyrate-producing bacteria, *Coprococcus comes* and *Eubacterium rectale*, which were associated with more anxious depression and less inhibitory control. Conflictingly, *Coprococcus* spp. and *Dialister* have been shown to be depleted in depressed patients in the literature [30].

## 5. Limitations

Human studies in general are complicated to design, as it is difficult to recruit patients who meet all inclusion and exclusion criteria, often resulting in a small sample [13,14,18,21]. In the case of longitudinal studies (especially if they last for some years) the main limitation is loss to follow-up, which again considerably reduces the sample size, as was the case in some reviewed studies [19,20]. There are two important key limitations to comparing gut microbiome studies: (i) variations in stool sample collection, preservation, DNA extraction, and bioinformatic analysis methods can impact the results; and (ii) the majority of studies only identify bacteria at the genus level, with very few analyses at the species level.

## 6. Conclusions and Recommendations

Here, we summarized the human literature to date, investigating the link between ELS and gut microbiome. We identified and described 13 studies of both pre- and postnatal stressors. An important observation is that few studies have used biomarkers of stress systems to complement self- or interview-reported stress. Furthermore, the types of early life stressors are very diverse, and results are generally weak (e.g., the level of phylogenetic resolution is family or at most genus), and so no firm conclusions can be drawn. Nonetheless, the review does demonstrate links between ELS and gut microbiome changes, with only 2 of the 13 studies not finding associations. Further research will be necessary to draw more robust conclusions. Moreover, to ensure consistency and comparability across studies, it would be beneficial to use the same validated questionnaires. Regarding the gut microbiome, future studies using standard procedures and species and strain resolution shotgun metagenomics sequencing are needed for analyses with a sufficient level of specificity. Finally, more attention should be paid to the influence of environmental variables (diet, physical activity, etc.) and sex on gut microbiome analysis.

## Figures and Tables

**Figure 1 nutrients-15-02566-f001:**
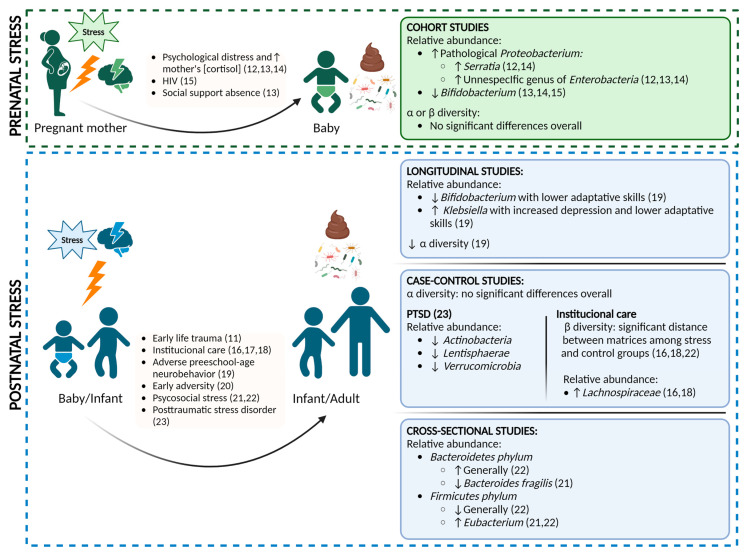
Overview of the main results compiled in the systematic review of prenatal [12,13,14,15] and postnatal stress studies [11,16,17,18,19,20,21,22,23].

**Figure 2 nutrients-15-02566-f002:**
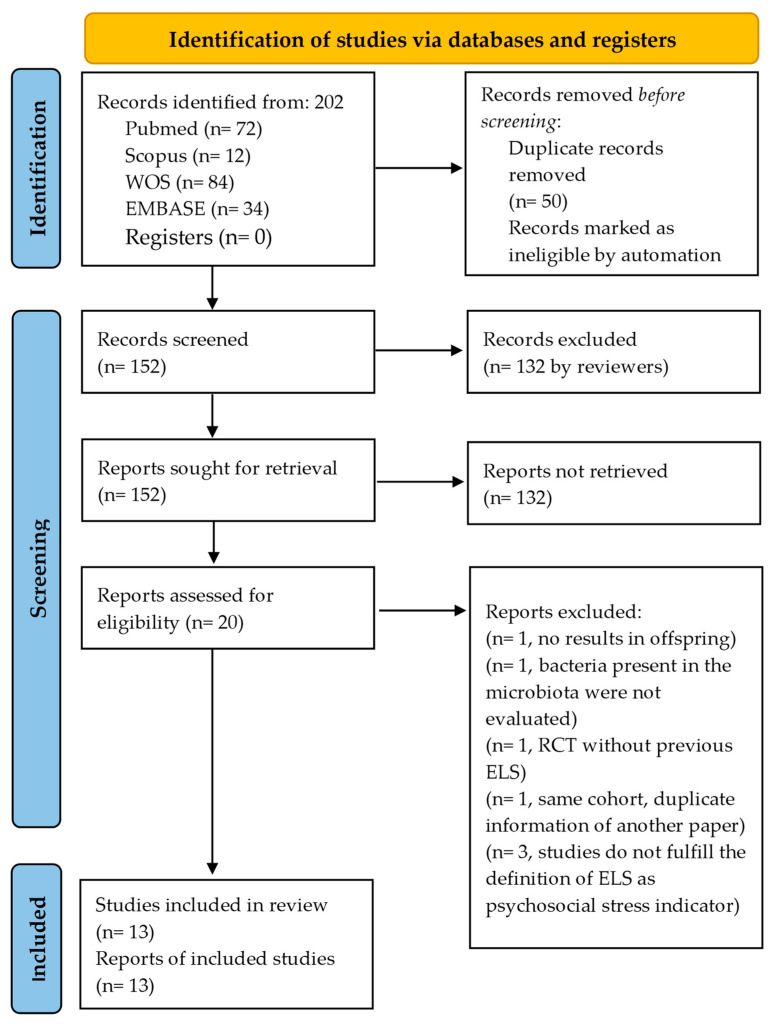
PRISMA flow diagram.

**Table 1 nutrients-15-02566-t001:** Quality assessment for the selected studies based on Newcastle-Ottawa Quality Assessment Scale for longitudinal cohort studies. A study can be awarded a maximum of one star for each numbered item within the Selection and Outcome categories. A maximum of two stars can be given for Comparability. Total score ranges from 0 to 9. Higher scores indicate better methodological quality.

Reference	Selection				Comparability	Outcome			Total
	Representativeness of Exposed Cohort	Selection Non-Exposed Cohort	Ascertainment of Exposure	Outcome of Interest Not Present at Start of Study	Comparability of Cohort	Assessment of the Outcome	Follow-Up Long Enough for Outcomes	Adequacy of Follow-Up of Cohorts	
Zijlmans et al., 2015 [14].	*	*		*	**	*	*		7
Aatsinki et al., 2020 [12].	*	*			**	*	*	*	7
Grant-Beurmann et al., 2022 [15].	*	*	*	*	*	*	*		7
Jahnke et al., 2021 [13].	*	*	*		**	*	*		7
Laue et al., 2021 [19].	*	*	*	*	**	*	*		8

**Table 2 nutrients-15-02566-t002:** Quality assessment for the selected studies based on Newcastle-Ottawa Quality Assessment Scale for case-control studies. A study can be awarded a maximum of one star for each numbered item within the Selection and Exposure categories. A maximum of two stars can be given for Comparability. Total score ranges from 0 to 9. Higher scores indicate better methodological quality.

Reference	Selection				Comparability	Exposure			Total
	Case Definition	Representativeness of the Cases	Selection of Controls	Definition of Controls	Comparability of Cases and Controls	Ascertainment of the Exposure	Same Method of Ascertainment	Non-Response Rate	
Malan-Muller et al., 2022 [23].	*	*	*	*	**	*	*		8
Hermes et al., 2020 [17].	*	*	*	*	**	*	*		8
Keskitalo et al., 2021 [20].	*	*	*	*	*	*	*		7
Callahan et al., 2020 [16].	*		*	*	**	*	*		7
Reid et al., 2021 [18].	*		*	*	**	*	*		7

**Table 3 nutrients-15-02566-t003:** Quality assessment for the selected studies based on a scale adapted from the Newcastle-Ottawa Quality Assessment Scale for cohort studies and an adapted Newcastle-Ottawa scale created for cross-sectional studies by Herzog et al., 2013 [44]. A study can be awarded a maximum of five stars in the Selection, two stars in the Comparability, three in the Outcomes and one in the Statistical test categories. Total score ranges from 0 to 9. Higher scores indicate better methodological quality.

Reference	Selection				Comparability	Outcome		Total
	Representativeness of the Sampler	Selection of the Control Group	Ascertainment of Exposure	Non-Respondents	The Subjects in Different Outcome Groups are Comparable	Ascertainment of the Outcome	Statistical Test	
Coley et al., 2021 [11].	*	*	**		**	**	*	9
Michels et al., 2019 [22].	*		**		**	**	*	8
Flannery et al., 2020 [21].	*		*		*	**	*	6

**Table 4 nutrients-15-02566-t004:** Characteristics and results of prenatal stress studies.

Study	Population		Design and Aim	Measures		Results/Outcomes
References	N	Characteristics		Microbiota	Stress	
Aatsinki et al., 2020 [12]	N = 446 Mother–infant pairs	- Pregnant women: - Recruitment at 14 gwk until 2.5 mths of baby’s life - Children: Age 2.5 mths - Country: Finland	Large prospective longitudinal study (FinnBrain Birth Cohort 2011–2015) Aim: to investigate the role of early life exposures (PPD and HCC) specifically on infant fecal microbiota	Infant fecal sample collection: at 2.5 mths Analysis of fecal samples: - DNA stool extraction (GXT Stool Extraction Kit VER 2.0 (Hain Lifescience GmbH, Nehren, Germany) - 16S rRNA gene sequencing (MiSeq, Illumina platform for the V4 region 16S rDNA)	Maternal hair samples cortisol measurement at 24 gwk (Cortisol Saliva kit): HCC Maternal prenatal psychological distress questionnaires, self-reported at 14, 24, and 34 gwk: (1) EPDS (2) SCL-90 (3) Daily Hassles (4) PRAQ-R2	Chronically elevated * maternal PPD symptoms were associated with proteobacteria in the infant: - Daily Hassles positively associated with: *Erwinia*, *Haemophilus*, and *Serratia*; - SCL positively associated with: *Campylobacter*, *Citrobacter*, and *Serratia*; - EPDS negatively associated with *Desulfovibrio* and positively with *Citrobacter* and *Serratia*; - PRAQ-R2 positively associated with - *Camplylobacter*, *Serratia*, and *Haemophilus*. Maternal chronic PPD * measures showed associations: - Positively: *Veillonella*, *Finegoldia*, *Dialister*, *Dorea*, and *Coprococcus*, *Actinomyces* and *Rothia*; - Negatively: *Akkermansia Pseudoramibacter*, *Phascolarctobacterium*, *Megamonas*, *Megasphera*, *Eubacterium*, *Epulopiscium*, *Anaerotruncus*, *Pseudoramibacter_Eubacterium*, *Paraprevotella*, *Parabacteroides*, *Odoribacter*, *Slackia*, *Actinobaculum*, and *Propionibacterium*. Maternal depression measures * (EDPS) showed associations: - Positively: *Butyricimonas* and *Prevotella* * Subjects scoring above the cut-off in two or more prenatal measurements (14, 24 and 34 gwk). Cortisol (hair) in mothers at 24 gwk was associated: - Negatively: *Slackia* and *Actinobaculum*, *Paraprevotella* and *Butyricimonas*, *Citrobacter*, *Ruminococcus*, Phascolarctobacter, Anaerotruncus, Enterococcus, and *Lactobacillus*. Neither maternal chronic PPD of approximately the past five mths of pregnancy nor HCC associated with infant fecal microbiota diversity.
Grant-Beurmann et al., 2022 [15]	N total = 272 mother–infant pairs N (HIV/HUU) = 131 N(HIV+/HEU) = 141	- Pregnant women with or without HIV infection Ages: 26–33 yrs (HUU) 31–37 yrs (HEU) - Babies born to these women: • Children HEU • Children HUU Study during first 18 mths of life - Country: Nigeria	Prospective cohort study of mother–infant pairs 2015–2018 Aim: to assess potential differences in the gut microbiota in infants born to HIV-positive and -negative mothers during the first 18 mths of life	Mother samples: - Vaginal swabs and stool (at 12 gwk and at birth) - Breast milk samples (at 6 wks and 6 mths postpartum) - Fecal sample (at enrollment and after birth) Infant fecal samples (meconium, 6 wks and 6, 9, 15 and 18 mths postpartum) Analysis of fecal and vaginal samples: - DNA extraction (DNeasy Power Soil Kit, Qiagen) - 16S rRNA gene sequencing (Illumina HiSeq 2500 modified for the V3–V4 region 16S rDNA) Analysis of metabolites in breast milk: UHPLC/MS/MS and library-based	Clinical assessment: Standardized questionnaires (medication and comorbidity history), general physical examination, anthropometric assessment	Maternal vaginal and infant fecal microbiota showed increased the diversity over time. Breastfeeding was associated with differences in gut microbiota of infants (HEU and HUU). The relative abundance association was: - Positively: *Bifidobacterium* and *Collinsella*; - Negatively: *Faecalibacterium* and *Streptococcus*. Bifidobacterium, significantly more abundant in the breastfeeding HUU infants than in breastfeeding HEU infants at 6 mths postpartum Low *Bifidobacterium* abundance associated with low weight Breast milk composition differed by time point and HIV infection status - 106 metabolites in the breast milk of mothers with or without HIV at 6 wks and 6 mths postpartum significantly differed between the two groups and time points; - Kynurenine and antiretroviral therapy more abundant in the breast milk of mothers with HIV; - Lower tiglyl carnitine (C5) and acylcarnitine concentrations in breast milk in mothers without HIV; - Nevirapine (antiretroviral therapy), high concentrations correlated with lower relative abundance of *Bifidobacterium longum* in cohort of 17 HEU infants.
Jahnke et al., 2021 [13]	N total = 25 mother–infant dyads	- Pregnant women Age: 18–50 Data collected over 12 mths Country: Ecuador (Galápagos, San Cristobal island)	Longitudinal cohort dyad study. Aim: to assess relationships among maternal precarity and HPA axis dysregulation during the peripartum period, infant gut microbiome composition, and infant HPA axis functioning	Infant fecal collection at 2 mths postpartum Analysis of fecal samples: - DNA stool extraction (Qiagen) - 16S rRNA gene sequencing (MiSeq, Illumina platform for the V4 region 16S rDNA)	Measures of maternal precarity taken during and after pregnancy: (1) Food insecurity assessed using the ELCSA (2) Chronic stress: PSS (3) Depression: measured by PHQ-8. (4) Social support: measured by PSS-Family and PSS-Friends scales Saliva sample (cortisol): - Maternal sample: at 34–36 wks of pregnancy and at 1 mth postpartum. Three samples were provided each time: (1) Immediately upon waking (CAR) (2) 30 min after waking (3) Prior to sleep - Infant samples collected (Salimetrics Infant Swabs and placed into Salimetrics Swab Storage) when the infants were: (1) 3 days: basal cortisol (2) 2 mths old: cortisol reactivity (before and 20–25 min after a stressor)	Gut microbiome alpha diversity: - Postpartum maternal depression, and high morning cortisol: significantly associated with lower Shannon diversity values in infant stool at 2 mths of age. Gut microbiome beta diversity: - Infants whose mothers were depressed at 2 mths postpartum clustered separately from infants whose mothers were not depressed (PCoA) and significant for UniFrac distance and Bray–Curtis dissimilarity; - Basal infant cortisol at 3 days old was significantly associated with beta diversity differences for weighted UniFrac distance and Bray–Curtis dissimilarity. Relative abundance association of predominant taxa: • Food insecurity associated with: - Higher relative abundance of *Proteobacteria* (an unspecified genus of *Enterobacteriaceae*); - Lower relative abundance of the family *Lachnospiraceae*. • Low friend support during pregnancy associated with: - Higher relative abundance of *Proteobacteria* (an unspecified genus of *Enterobacteriaceae*). • Postpartum depression associated with: - Lower relative abundance of *Actinobacteria* (genus *Bifidobacterium*). - Lower abundance of the family *Lachnospiraceae*; - Higher relative abundance of the family *Streptococcaceae.* • Low family support during pregnancy associated with: - Higher relative abundance of genus *Bifidobacterium*; • Low family support in the postpartum associated with: - Lower abundance of *Bifidobacterias.* • Low maternal CAR during pregnancy associated with: - Lower abundance of *Bacteroidetes* and • Low maternal CAR postpartum associated with: - Higher abundance *Bacteroidetes* and *Veillonella* • High postpartum morning cortisol associated with: - Greater relative abundance of *Bacteroidetes* (*Bacteroides*) • High infant basal cortisol at 3 days postpartum associated with: - Lower abundance of *Actinobacteria* (genus *Bifidobacterium*) - Higher abundance of *Proteobacteria* (an unspecified genus *Enterobacteriaceae*).
Zijlmans et al., 2015 [14]	N total = 56 children	Children followed from the 3rd trimester of pregnancy until 110 days after birth Children: - Healthy - Born at full term - (≥37 wks) - Vaginal delivery - 5-min APGAR score ≥ 7 Country: The Netherlands	Prospective longitudinal study Aim: to investigate the development of the gut microbiota as a potential pathway linking maternal prenatal stress and infant health	Infant fecal sample collection: 5 time points (from birth until ± 110 days of life): T1–T5 Analysis of fecal samples: - DNA stool extraction (repeated bead beaten method). - 16S rRNA gene directed sequencing (V1 and V6 phylogenetic Microarray–HITChip)	Questionnaires (3rd Trimester): (1) STAI: anxiety (2) PRAQ-R (3) APL: daily hassles (4) PES: pregnancy-related daily hassles Maternal prenatal cortisol: 5 saliva samples (2 consecutive days) Mothers report infant gastrointestinal symptoms and allergic reactions monthly using the ICPC	Sum of: stress questionnaire scores + cortisol concentration (12 a.m.) were most strongly associated with the infant microbiota Infants with high cumulative stress (high reported stress + high cortisol saliva measure) showed: - Higher relative abundance of Proteobacterial groups known to contain pathogens (*Escherichia*, *Serratia*, and *Enterobacter*); - Lower relative abundances of lactic acid bacteria (i.e., *Lactobacillus*, *Lactoccus*), *Aerococcus*, and *Bifidobacterium*; - More gastrointestinal symptoms and allergic reactions reported by the mothers, suggesting an association with the aberrant microbiota.

APL: Alledaagse Problemen Lijst. APGAR score: a score assigned to a newborn baby to evaluate its physical condition at birth. CAR: cortisol awakening response. EDPS: Edinburgh Postnatal Depression Scale. ELCSA: Latin American and Caribbean Food Security Scale, a tool used to assess food insecurity; gwk: gestational week. HCC: hair cortisol concentration. HEU: HIV-exposed uninfected children. HIV: Human Immunodeficiency Virus. HITChip: Human Intestinal Tract Chip, a type of phylogenetic microarray specifically designed to study the gut microbiota. HUU: HIV-unexposed uninfected children. HPA: Hypothalamic-pituitary-adrenal. ICPC: International Classification of Primary Care MiSeq: Illumina sequencing platform. Mths: months. N: Sample size. PES: Pregnancy Experience Scale. PHQ-8: Patient Health Questionnaire-8. PPD: maternal–prenatal psychological distress. PRAQ-R: the pregnancy-related anxieties questionnaire-revised. PRAQ-R2: Pregnancy-Related worries and anxiety related with the newborn. PSS: Perceived Stress Scale. PSS-Family: Perceived Social Support-Family scale. PSS-Friends: Perceived Social Support-Friends scale. rRNA: Ribosomal ribonucleic acid. SCL-90: Symptom Checklist-90, anxiety subscale. STAI: State-Trait Anxiety Inventory. UHPLC/MS/MS: Ultra High-Performance Liquid Chromatography/Mass Spectrometry/Tandem Mass Spectrometry. V1–V6: Regions of ribosomal DNA sequence. wks: weeks. yrs: years.

**Table 5 nutrients-15-02566-t005:** Characteristics and results of postnatal stress studies.

Study	Population		Design	Measures		Results/Effect on Outcomes
References	N	Characteristics		Microbiota	Stress	
Longitudinal Studies						
Laue et al., 2022 [19]	N = 260 children from mothers in NHBCS cohort with complete stool and BASC data	Male = 144 Female = 116 Mean gestational age 39.5 wks C-section *n* = 78 Country: U.S.A	NHBCS longitudinal pregnancy cohort Aim: to identify sex-specific prospective associations between the early life gut microbiome and preschool-age neurobehavior	Stool sampling in children at 6 wks, 1 yr and 2 yr postpartum Analysis of fecal samples: - DNA extraction (Zymo Fecal DNA extraction kit) - 16SrRNA sequencing (MiSeq, Illumina platform for the V4 and V5 region 16S rDNA) - smaller set of samples at 6 wk (*n* = 97) and 1 yr (*n* = 98) underwent shotgun metagenomic sequencing (Next Illumina platform) but not clear how sample selection was made	Questionnaires: Behavioral development assessed by parent using BASC-2 at age 3 with 10 subscales scores: - Anxiety - Depression - Internalizing Problems - Attention Problems - Hyperactivity - Behavioral Symptoms Index - Externalizing Problems - Developmental Social Disorders - Social Skills - Adaptive Skills behaviors	Alpha diversity: - Higher diversity at 6 wks was related to lower depression in the overall sample and with lower anxiety and Internalizing composite scale scores in boys; - No significant associations at 1 yr; - At 2 yrs, increased diversity associated with better developmental social scores, social skills, and adaptive skills in boys, but with worse scores on these scales in girls. Beta diversity: - Overall, not related to BASC2 scale in fully adjusted models at any time point (but some existed in smaller models). In boys, significant differences in beta diversity at 6 wks occurred with Anxiety scores (direction of effect not shown), but were not significant when test for sex differences. Microbiome taxa: Boys at 6 wks: Adaptive skill composite score linked to higher abundance of: - *Bifidobacterium*, *Bacteroides vulgatus*, *Streptococcus*. Worse adaptive skills in boys linked to: - *Klebsiella*, *Clostridium* and *Haemophilus*. Better depression scores linked to: - *Tyzzerella nexelis*. Overall, stool sample 1 yr: Worse hyperactivity scores linked to relative abundance of *Faecalitalea*. Overall, stool sample at 2 yrs: 4 *Blautia* ASVs linked to worse measures of Hyperactivity. Metagenomic species related to BASC-2 scores: 6 wks: - No bacterial taxa were strongly associated with any BASC-2 scale; there was a poor association between *Eggerthella lenta* and Depression; - *Klebsiella oxytoca* was adversely associated with Adaptive Skills Composite scores in boys; - *Granulicatella* was associated with worse Anxiety scores in girls. 1 yr: No associations significant at the FDR level observed at 1 yr, except for lower Depression and Internalizing problems score in girls with abundance of *Streptococcus peroris*. Bacterial functional pathways related to BASC-2 scores: - Depression scores: associated with relative abundance of microbial functional pathways at 1 yr in boys, including PWY0-845, the super pathway of L-aspartate and L-asparagine biosynthesis, and L-ornithine biosynthesis II; - Several pathways whose relative abundances were associated with better Depression scores in boys related to vitamin B6 biosynthesis or salvage (PWY0-845 at 1 yr, PYRIDOXSYN-PWY at 1 year, and PWY-7204 at 6 wks); - A strong correlation between PWY0-845 and PYRIDOXSYN-PWY may be responsible for the concordant results, but PWY-7204 was not correlated with either of the other pathways
Case-control studies						
Callahan et al., 2020 [16]. (2nd study)	N total = 16. N Controls = 8 N children with EA *n* = 8	Controls: - Female *n* = 7 - Mean age = 11 yrs EA: - Female *n* = 5 - Mean age = 13 yrs Country: U.S.A	Aim: to study association between early caregiving adversity, the gastrointestinal (GI) microbiome, and brain reactivity to threat stimuli (fear faces)	Single fecal sample collection. Analysis of fecal samples: - DNA extraction (RNA Power Siol kit, Mo Bio Laboratories). - 16S rRNA (MiSeq, Illumina platform for the V1 and V2 region 16S rDNA)	EA group, i.e., institutional or foster care followed by international adoption vs. controls	Alpha Diversity: - EA had lower counts (lower richness) of bacteria but no differences were observed in the relative abundance of those bacteria (Shannon index). Beta Diversity: - Caregiving group (COMP) accounted significantly for variation in bacterial distance matrices between Individuals. Two biomarkers were identified, from the order Clostridiales: one was an unknown genus in the family *Lachnospiraceae*, and the other was from an unknown genus and family. Both biomarkers were higher in children from the COMP than EA groups.
Hermes et al., 2020 [17]	N = 98 infants CC infants HOME infants	CC group: - Females = 20 - Mean age: PRE = 87.8 days POST = 118.4 days - Birth weight = 3630 g - C-section *n* = 6 HOME group: - Females = 24 - Mean age: PRE = 76.7 days POST = 106.5 days - Birth weight = 3636 g - C-section = 3 Country: The Netherlands	Case-control study. CC entry at 3 mths vs. HOME Aim: to investigate whether CC, as compared with being cared for by the parents at home, alters the composition of the gut microbiota	Stool samples collected at 10 wks post-partum and 4 wks after CC entrance. Analysis of fecal samples: - DNA stool extraction (repeated bead beaten method). - 16S rRNA gene directed sequencing (V1 and V6 phylogenetic microarray, HITChip)	CC vs. HOME	No significant effect of CC entry or the number of half-days in CC compared with staying at home, on the microbiota using Redundancy analysis, Random Forest, and Bayesian linear models.
Keskitalo et al., 2021 [20]	N = 193 2.5-mth-old infants with both fecal sample and salivary stress response measurement	Males = 102 Females = 91 Vaginal birth = 154 Mean age = 10.4 wks Mean birth weight = 3616 g Gestational age mean = 40 wks Country: Finland	Nested case-control study with infants exposed to different types of maternal prenatal psychological stress vs. non-exposed controls (Subset of FinnBrain Focus cohort study) Aim: to identify the potential link between the cortisol stress response and the gut microbiota at the age of 2.5 mths	Single fecal sample collected (2.5 mths) Analysis of fecal samples: - DNA extraction (GTX Stool extraction kit VER 2.0), - 16S rRNA gene sequencing (Illumina MiSeq Platform for the V4 region 16S rDNA)	Infants exposed to maternal prenatal psychological stress vs. non-exposed controls Stress test: - 2.5 mths of age - Standard pediatric exam (saliva sampling at 0, 15, 25 and 35 min after stressor). Assayed with Cortisol Saliva Luminescence Immunoassay. AUCi calculated	Post-stressor salivary cortisol AUCi negatively associated with alpha diversity but not beta diversity. No evident associations between cortisol stress response and bacterial taxa when ALDEx2 analyses were adjusted for the selected covariates.
Malan-Muller et al., 2022 [23]	N total = 137 N PTSD = 79 PTSD N TECs = 58	Age PTSD = 32–52 yrs Age TECs = 38–58 yrs Female PTSD = 63 Female TECs = 47 Country: South Africa	Case-control study to investigate associations between the gut microbiome and mental health outcomes PTSD and TECs	Stool samples collected the same wk as the clinical assessment. Analysis of fecal samples: - DNA stool extraction - 16S rRNA (Illumina Miseq Platform for the V3-V4 region 16S rDNA)	PTSD vs. TE To evaluate PTSD: - Diagnostic and Statistical DSM-5 to select PTSD or TECs - CAPS-5: to diagnose PTSD - CAPS-5: 30-item structured interview to diagnose lifetime and assesses symptoms of PTSD - CTQ - International Neuropsychiatric Interview [MINI] version 6.0: to diagnose MDD and anxiety disorders - LEC-5: to evaluate potentially traumatic lifetime events	Mental outcome - Individuals with PTSD: ↑ CAPS-5 and CTQ scores; ↑ Psychotropic medication use; ↑ Prevalence of MDD and anxiety disorders. Alpha diversity: - No differences in gut microbiome community composition between PTSD and TECs (genus or phylum level); - Psychotropics influenced the fecal community composition (genus and phylum level). Taxonomic composition: Associations between traumatic experiences, psychiatric diagnosis and taxonomic abundance: - MDD participants had a significantly higher relative abundance of the phylum Bacteroidetes. - Genera higher in individuals with PTSD compared with TECs (positively correlated with CAPS-5 score): - *Mitsuokella* - *Odoribacter* - *Catenibacterium* - *Olsenella* Association between psychotropic medication use and taxonomic abundance: - Individuals using psychotropic medication at the time of sampling: ↑ *Ruminococcus* rel. abundance ↓ *Akkermansia* rel. abundance positively associated with: *Bacteroidetes* *Firmicutes* negatively associated with: *Verrucomicrobia*
Reid et al., 2021 [18]	N total = 38 N PI = 17 N COMP = 18	PI Mean age = 16.8 yrs Female = 13 CMV Seropositive = 14 COMP Mean age = 15.9 yrs Female = 11 CMV seropositive = 5 Country: U.S.A	Case-control study to assess the relation between ELS in PI participants: gut microbiome and inflammatory markers PI participants spent 70% of preadoption life in institutional care COMP participants were from families with comparable education and incomes Parent/youth questionnaire report of health and diet Stool and blood samples were collected once	Fecal samples were collected at home Analysis of fecal samples: - DNA stool extraction by bead beating (BioSpec) and phenol:choloroform extraction. - 16S rRNA (Illumina MiSeq platform for the V3 region 16S rDNA)	PI vs. COMP There was no questionnaire for evaluating stress However, PI participants spent 70% of preadoption life in institutional care, considered as an early life stress factor in the study	Alpha diversity: - No significant group or sex differences Beta diversity: - Significant differences between COMP and PI individuals. - Negative binomial regression analysis revealed OTUs significantly increased in PI group over the COMP group: - *Prevotella* - *Bacteroides* (*Bacteroidaceae*) - *Coprococcu* - *Streptococcus* - *Escherichia* ↑ CMV seropositivity in PI - OTUs different in CMV seropositive and CMV seronegative: - *Escherichia* - *Ruminococcaceae* - *Lachnospiraceae* - *Catabacteriaceae* - 37 OTUs significantly different between COMP and PI; - Significant association between CMV seropositivity and *Escherichia* was related to seropositive individuals in the PI group.
Cross-sectional studies						
Coley et al., 2021 [11]	Ntotal = 128 healthy adults	Male *n* = 43 Female *n* = 85 Age females ≤ 45 years and premenopausal groups based on ETI-SR score - High ETI: mean age (SD) = 28.81 - Low ETI mean age (SD) = 27.28 Country = U.S.A	Case-control based on ETI score (ETI-SR total > 4) Aim: to test whether ELS-related alterations in gut microbial metabolites are associated with alterations in brain connectivity, disordered mood, and increased vulnerability to stress in adulthood	Single fecal sample collected Analysis of fecal samples: - DNA extraction with bead beating (Qiagen Powersoil DNA isolation kit) - 16S rRNA sequencing (illumina HiSeq 2500 for the V4 region 16S rDNA)	ETI score > 4 vs. ETI score ≤ 4 Questionnaires: - ETI-SR to evaluate ELS, 27 items, 4 domains: - General trauma - Physical punishment - Emotional abuse - Sexual abuse	No significant relationships between history of ELS exposure and microbial alpha diversity, microbial beta diversity, or relative taxonomic abundance, at either phylum or genus level. Symptoms of anxiety, depression and body mass index correlated significantly with feces metabolites: urate, glutamate gamma-methyl ester, and 5-oxoproline.
Flannery et al., 2020 [21]	N = 40 children from a larger study	Mean age = 6.12 yrs (range 5–7). Female = 23 Country: U.S.A	Subsample from a midsize city in the Pacific Northwest of the U.S.A. already participating in a larger study of families, asked to participate in FU to collect stool samples (Cross-sectional study) Aim: determine how both the microbial taxa and the specific genetic functions they encode associate with subclinical child behavioral dysregulation symptoms (hereafter “behavioral dysregulation”), socioeconomic risk, and caregiver behavior	Single fecal sample collection Analysis of fecal samples: - DNA extraction with MoBio PowerLyzer PowerSoil kit. - Shotgun metagenomic analyses using Illumina GAIIx platform, taxonomically profiled using MetaPhlAn, metabolically profiled by HUMAnN, and assembled for gene annotation and clustering into a unique catalogue	Questionnaires: - CBQ - CBCL - PSI - IEMP - FiveFactor Mindfulness Questionnaire Life Event Checklist as index for adverse home environment exposure Other questionnaires: Gut history questionnaire filled out by parent	Significant association between functional composition and impulsivity. *Bacteroides fragilis* associated with reduction in: - Aggressively; - Emotional reactivity; - Sadness; - Impulsivity; - Lower family incidents (total score on LEC) test). *Coprococcus* comes and *Eubacterium rectale* associated with increase in: - Anxious depression; - Less inhibitory control. *Roseburia inulinivorans* associated with reduced: - Depressive problems. Interactions between the gut microbiome, socioeconomic risk, and behavioral dysregulation. Associations involving: - Monoamine metabolism (tryptophan, tyrosine, glutamate, and leucine) and microbehost antagonism (types II, III, and VI secretion systems).
Michels et al., 2019 [22]	N total = 93 children/adolescents	Age = 8–16 yrs Country: Belgium	Cross-sectional study to investigate associations between gut microbiome and psychosocial stress in children/adolescents Children and adolescents recruited for the longitudinal ChiBS study (Michels et al., 2012)	Unique fecal sample: collected at home Analysis of fecal samples: - DNA stool extraction using a Lysis Buffer and glass beads for FastPrep and phenol:choloroform extraction. - 16S rDNA gene sequencing, Illumina Miseq Platform for the V3-V4 region 16S rDNA	Stress markers: (1) hair cortisol levels by chromatography coupled to tandem mass spectrometry (AB Sciex 5500 triple quadrupole) (2) Pnn50-parasympathetic activity: - 10 min - RR-intervals recorded at a sampling rate of 1000 Hz with an elastic electrode Questionnaires: (1) Negative events: - Self report - CLES-C 36-item (2) Negative and positive emotions: - Self-report - The feelings happy, anger, anxiety and sadness were rated from 0 to 10. Anger+ Anxiety + Sadness = ‘Negative emotions’ (3) Emotional problems: - Reported by the parents - Standardized ‘Strengths and Difficulties Questionnaire’ for their child - 5 items were used	Stress biomarkers: High stress: low pnn50 and more negative events; High pnn50: positively related to happiness. Alpha diversity: - General (8–16 yrs): High stress (more negative events and low pnn50) was significantly related to lower diversity (Simpson Index). - Preadolescents: same observations with additionally negative events and less observed species. - Adolescents: no significant differences in alpha diversity by stress were found. Beta diversity: - General (8–16 yrs): for pnn50 and happiness there was a significant separation between high and low stress groups; - Preadolescents: happiness and pnn50 showed significant microbiome separation; - Adolescents: no significant microbiome separation. Phylogenetic differences: - Pnn50 and happiness provided the clearest distinction in microbiota composition; - Low pnn50 or happiness (high stress) associated with: ↓ Firmicutes relative abundance, mainly Clostridiales (most often genera under the family of Lachnospiraceae and Ruminococcaceae); ↓ Bacteroidetes (Bacteroidales); ↓ Euryarchaeota (Methanobrevibacter). Taken all stress measures together: - 35 genera appeared distinctive for stress; - The genera with greatest effect sizes were: - *Ruminococcaceae UCG014*; - *Bacteroides*; - *Phascolarctobacterium*; - *Tenericutes uncultured*; - *Eubacterium coprostanoligenes*; - *Methobrevibacter*; - *Blautia*. - As a general pattern: High stress was associated with - Lower Firmicutes (phylum level); - Higher *Bacteroides*, *Parabacteroides*, *Rhodococcus*, *Methanobrevibacter* and *Roseburia*; - Lower *Phascolarctobacterium* (genus level). Conflicting results between different stress measures: *Ruminococcaceae*, *Tenericutes*, *Eubacterium coprostanoligenes*, *Prevotella* and *Christensenellaceae*. Differential results in preadolescents vs. adolescents were also evident. - Adjusted and unadjusted taxonomic differences were also more pronounced for happiness and pnn50, being associated respectively with 24 OTUs (11.8% of bacterial counts) and 31 OTUs (13.0%).

AUCi: Area Under the Curve with interpolation. BASC-2: Behavior Assessment System for Children BASC-2. CC: center-based childcare infants. CBCL: Child Behavior Checklist. CBQ: child behavior questionnaire. CTQ: Childhood Trauma Questionnaire. CAPS-5: Clinician Administered Posttraumatic Stress Disorder Scale for DSM–5. CLES-C: Coddington Life Events Scale for Children. COMP: youth reared in birth families. CMV: cytomegalovirus. EA: early adverse. ETI-SR: Early Traumatic Inventory-Self report. HOME: control infants cared for at home by parents. HITChip: Human Intestinal Tract Chip. IEM-P: Interpersonal Mindfulness in Parenting. LEC: Life Event Checklist. LEC-5: Life Events Checklist from the DSM-5. DMS-5: Manual of Mental Disorders, Fifth Edition. NHBCS: New Hampshire Birth Cohort Study. PI: youth internationally adopted and previously institutionalized. PNN50: Heart Rate Variability. PSI: Parenting Stress Index. PTSD: posttraumatic stress disorder. TECs: trauma-exposed controls. wks: weeks. yrs: years.

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
