# Peer review of "The Gut Microbiome in Early Life Stress: A Systematic Review"

_nutrients, 2023, doi:10.3390/nu15112566_

Round 1
Reviewer 1 Report
In this systematic review, the authors have summarized the literature to date on human 598 studies on the link between ELS and gut microbiome. I think it is of general interest to the readers, thus I would recommend it for publication if the following issues can be addressed.
1. Uniform all the references, especially the initial letters of titles.
2. et al in the articles should be italic.
Author Response
Tuesday, May 2, 2023
Dear Reviewer,
I am writing to inform you that I have implemented the changes you suggested in my document. With regards to Reviewer 1's comments, I have carefully reviewed and revised all references to ensure consistency throughout the document. As a result, I did not highlight any specific changes as they have all been addressed. Additionally, I have italicized all occurrences of "et al.", all of them are highlighted in green.
Thank you for your valuable feedback, and please let me know if you have any further suggestions.
Kind regards,
Ana Agustí

Reviewer 2 Report
Comments to nutrients-2373194
The gut microbiome in early-life stress: a systematic review
General concerns:
this review is focused on the evaluation of the influence of early-life stress on gut microbiome in babies and children, and considered both pre-natal and post-natal stresses.
The authors carefully and scrupulously reviewed the recent literature and selected 13 papers between 2015 and 2022. The criteria for inclusion or exclusion of the works found have been well clarified and made explicit. The 13 selected studies were split according to whether they treated prenatal or postnatal stress.
Microbiome data was extrapolated from each study and then compared and summarized in the discussion.
The authors also highlighted the difficulty of finding human studies that could be comparable, for various factors such as e.g. the diversity of the questions in the questionnaires, the different methods of collecting stool samples and bioinformatics analysis and processing with which the microorganisms of the microbiota are identified, etc. Furthermore, in longitudinal studies, a limiting factor is also the constancy and reliability of the recruited patients.
The work is well understandable and written in good English.
In my opinion the article can be published with minor revisions.
I have noticed some minor points:
-the bibliographic references are not listed in ascending order of numbering, for example, the last reference of section 1 (introduction) is n. [34]; in section 2 (Materials and Methods) the first reference is the n. [51] (line 106), then followed by no. [42] (line 142) and [43] (line 148). It is necessary to rearrange the numbering of the references according to the order of citation.
Line 108: “The search to identifying the published articles” correct with: “The search to identify the published articles”
Line 213: table 1, “Comparability of cohort…”, remove the ellipsis
Lines 301-304: remove the bold and change the formatting as required by the journal style
Line 305: format table 4 as table 5 with vertical lines separating the columns because it is not well understood
Lines 381-413: remove the bold and change the formatting as required by the journal style
Lines 644, 648 and other in Reference section: check the boldness and style of the references, it is not required to report the month of publication before the volume and pages
Line 669: “, ecal Microbiota Transplantation (FMT)” correct with “, Fecal Microbiota Transplantation (FMT)”
Author Response
Tuesday, May 2, 2023
Dear Reviewer,
I am writing to inform you that I have implemented the changes you suggested in my document. Regarding Reviewer 2's feedback, I have rearranged the numbering of references to match the order of citation. The errors detected since reference number 35 have been corrected and I have highlighted in yellow the first time a new citation appeared. I have also made several specific changes, such as correcting the sentence in line 108 to read, "The search to identify published articles." In the lines 301-304 I removed the bold and changed the formatting as required by the journal style. In the line 305 I have included lines in the format table 4 separating the columns. In the lines 381-413 I removed the bold and changed the formatting as required by the journal style. In the lines 644, 648 and the rest of Reference section I checked the boldness and style of the references I have removed the month of publication before the volume and pages. Finally, in the line 669 I corrected “ecal Microbiota Transplantation (FMT)” to “Fecal Microbiota Transplantation (FMT)”. All these changes have been highlighted in yellow.
Thank you for your valuable feedback, and please let me know if you have any further suggestions.
Kind regards,
Ana Agustí
